# Semi-Supervised Semantic Segmentation via Adaptive Equalization Learning

**Hanzhe Hu[1,4]*    Fangyun Wei[2]†    Han Hu[2]    Qiwei Ye[2]    Jinshi Cui[1]    Liwei Wang[1,3]†**

[1]Key Laboratory of Machine Perception (MOE), School of EECS, Peking University
[2]Microsoft Research Asia
[3]Institute for Artificial Intelligence, Peking University
[4]Zhejiang Lab
huhz@pku.edu.cn {fawe, hanhu, qiwye}@microsoft.com
{cjs, wanglw}@cis.pku.edu.cn

## Abstract

Due to the limited and even imbalanced data, semi-supervised semantic segmentation tends to have poor performance on some certain categories, e.g., tailed categories in Cityscapes dataset which exhibits a long-tailed label distribution. Existing approaches almost all neglect this problem, and treat categories equally. Some popular approaches such as consistency regularization or pseudo-labeling may even harm the learning of under-performing categories, that the predictions or pseudo labels of these categories could be too inaccurate to guide the learning on the unlabeled data. In this paper, we look into this problem, and propose a novel framework for semi-supervised semantic segmentation, named adaptive equalization learning (AEL). AEL adaptively balances the training of well and badly performed categories, with a confidence bank to dynamically track category-wise performance during training. The confidence bank is leveraged as an indicator to tilt training towards under-performing categories, instantiated in three strategies: 1) adaptive Copy-Paste and CutMix data augmentation approaches which give more chance for under-performing categories to be copied or cut; 2) an adaptive data sampling approach to encourage pixels from under-performing category to be sampled; 3) a simple yet effective re-weighting method to alleviate the training noise raised by pseudo-labeling. Experimentally, AEL outperforms the state-of-the-art methods by a large margin on the Cityscapes and Pascal VOC benchmarks under various data partition protocols. Code is available at https://github.com/hzhupku/SemiSeg-AEL.

## 1    Introduction

Supervised semantic segmentation requires pixel-level labeling, which is expensive and time-consuming. This paper is interested in semi-supervised semantic segmentation, which can greatly reduce the efforts of pixel-level annotation, yet may maintain reasonably high accuracy. One problem of common semantic segmentation datasets is that the pixel categories tend to be imbalanced, e.g., the pixel amount of head classes can be hundreds of times larger than that of tailed classes in the widely used Cityscapes dataset [1]. The situation is more serious in the semi-supervised setting where tailed classes may have extremely few samples. We note that recent approaches are mainly dedicated to the design of consistency regularization [2, 3, 4, 5, 6, 7] and pseudo-labeling [8], almost all of which neglect the imbalance problem and treat each category equally, leading to a biased training. These approaches may even harm the learning of tailed classes, as inaccurate predictions or pseudo labels of under-performing categories could falsely guide the learning on unlabeled data.

---

* This work is done when Hanzhe Hu was an intern in MSRA. † Corresponding author.

35th Conference on Neural Information Processing Systems (NeurIPS 2021).

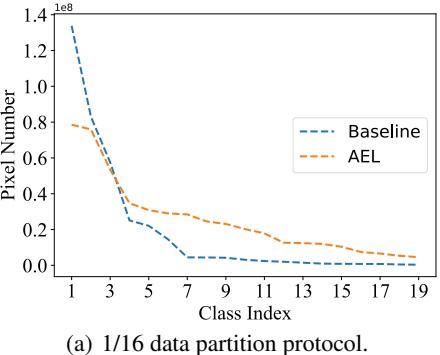

(a) 1/16 data partition protocol.

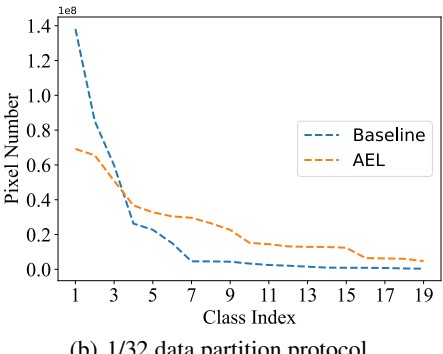

(b) 1/32 data partition protocol.

Figure 1: We count the the training samples of each category on Cityscapes `train` set under 1/16 and 1/32 data partition protocols, and compare the proposed AEL with a strong semi-supervised learning baseline described in Section 3.2 which treats each category equally. Our method strives to tilt training towards tailed categories which usually tend to be under-performing.

This paper aims to alleviate this biased training problem. We propose a novel Adaptive Equalization Learning (AEL) framework, which adaptively balance the training of different categories as shown in Figure 1. Our design follows two main principles: 1) increasing the proportion of training samples from the under-performing categories; 2) tilting training towards under-performing categories. Concretely, we maintain a confidence bank to dynamically record the category-wise performance at each training step, which indicates the current performance of each category. Following principle 1), we propose two data augmentation approaches named adaptive Copy-Paste and adaptive CutMix, which give more chance for under-performing categories to be copied or cut. Following principle 2), we present an adaptive equalization sampling strategy to encourage pixels from under-performing categories to be sufficiently trained. In addition, we also introduce a simple yet effective re-weighting strategy which takes the model predictions into account to alleviate the issue that semi-supervised learning usually suffers from the training noise.

Experimentally, by using the DeepLabv3+ with ResNet-101 backbone, the proposed AEL outperforms state-of-the-art methods by a large margin on the Cityscapes and PASCAL VOC 2012 benchmarks under various data partition protocols. Specifically, it achieves 74.28%, 75.83% and 77.90% on Cityscapes dataset under 1/32, 1/16 and 1/8 protocols, which is +16.39%, +12.87% and +8.09% better than the supervised baseline. When evaluated on PASCAL VOC 2012 benchmark, it achieves 76.97%, 77.20% and 77.57% under 1/32, 1/16 and 1/8 protocols, which is +6.83%, +6.60% and +4.45% better than the supervised baseline. Moreover, the proposed approach also proves to improve the segmentation model trained on the full Cityscapes `train` set by +1.03% by leveraging $5,000$ images from the Cityscapes `coarse` set as unlabeled data, achieving 81.95%.

## 2 Related Work

**Semi-Supervised Learning.** Recent years have witnessed a significant progress in the SSL field. Most of them can be categorized into consistency regularization, entropy minimization [9] and pseudo-labeling. Consistency regularization [10, 11, 12] enforces consistency in predictions between different views of unlabeled data. Pseudo-labeling [13, 14] trains the model on the unlabeled data with pseudo labels generated from the model's own predictions. Furthermore, [10, 15, 16, 17, 18] use a low softmax temperature to sharpen the predictions of unlabeled set. Our method refers to Mean Teacher [11] and FixMatch [14] when designing our basic framework.

**Semi-Supervised Semantic Segmentation.** Existing semi-supervised semantic segmentation methods mainly focus on the design of consistency regularization and pseudo-labeling. Cutmix-Seg [2] applies CutMix augmentation on the unlabeled data. CCT [4] introduces a feature-level perturbation and enforces consistency among the predictions of different decoders. GCT [19] performs network perturbation by using two differently initialized segmentation models and encourages consistency between the predictions from the two models. PseudoSeg [8] focuses on improving the quality of

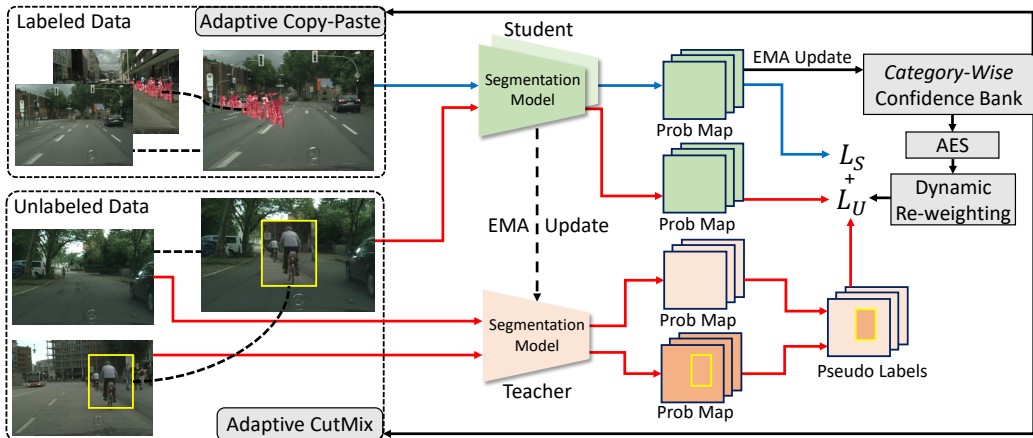

Figure 2: Overview of AEL. We adopt the teacher-student architecture as our basic framework. The teacher model is updated by the exponential moving average (EMA) of the student model. Confidence bank is uesd to dynamically record the category-wise performance during training. Adaptive CutMix and adaptive Copy-Paste are applied on the unlabeled and labeled data respectively to provide sufficient training samples from the under-performing categories. Adaptive equalization sampling (AES) encourages the training to involve more samples from the under-performing categories to make the training unbiased. Dynamic re-weighting strategy aims to alleviate the noise of pseudo-labeling.

pseudo labels. Though achieving satisfactory improvements over the supervised baseline, none of the aforementioned methods explore the biased learning issue in semi-supervised semantic segmentation.

**Class Imbalance in Semi-Supervised Learning.** Although SSL has been extensively studied, class imbalance problem in SSL is relatively under-explored, especially for semantic segmentation. Yang *et al.* [20] demonstrate that leveraging unlabeled data can alleviate imbalance issue. Hyun *et al.* [21] propose a suppressed consistency loss for class-imbalanced image classification problems. CReST [22] introduces a self-training framework for imbalanced SSL. Our method, though not explicitly targeting at the class imbalance problem, focuses on improving the performance of under-performing categories which are mostly tailed classes. Moreover, we refer to the ideas of re-sampling [23, 24] and re-weighting [25, 26], which are designed for class imbalance problem.

## 3 Method

Given a labeled set $\mathcal{D}^l = \{(\boldsymbol{x}_i^l, \boldsymbol{y}_i^l)\}$ and an unlabeled set $\mathcal{D}^u = \{\boldsymbol{x}_i^u\}$, the objective of semi-supervised semantic segmentation is to learn a segmentation model by efficiently leveraging both labeled and unlabeled data. In this section, we first present an overview of the proposed AEL in Section 3.1. Then we describe our basic framework for semi-supervised semantic segmentation in Section 3.2. Finally, the details of AEL are introduced in Section 3.3.

### 3.1 Overview

Figure 2 displays an overview of AEL, which is a data-efficient framework for semi-supervised semantic segmentation. It is composed of two parts: 1) a basic framework which contains a teacher model for pseudo-labeling and a student model for online learning; 2) dedicated modules which encourages the under-performing categories to be sufficiently trained by effectively leveraging both labeled and unlabeled data. We use the proposed confidence bank to dynamically record the category-wise performance during training, and thus we can easily identify which categories are not sufficiently trained. For those unsatisfactory categories, we present two data augmentation methods to increase their frequency of occurrence in a training batch, namely adaptive CutMix which is applied on the unlabeled data, and adaptive Copy-Paste which is applied on the labeled data. To make the model towards the unbiased learning, we propose the adaptive equalization sampling and dynamic re-weighting strategies to involve enough samples from the under-performing categories into the training, and alleviate the noise raised by pseudo-labeling simultaneously.

## 3.2 Basic Framework

We first set up a basic framework for semi-supervised semantic segmentation. The framework consists of a student model and a teacher model. The teacher model has the same architecture as the student model, but uses a different set of weights which are updated by exponential moving average (EMA) of the student model [11]. Following FixMatch [14], we use the teacher model to generate a set of pseudo labels $\hat{\mathcal{Y}} = \{\hat{y}_i\}$ on the weakly augmented unlabeled data $\mathcal{D}^u$. Subsequently, the student model is trained on both labeled data $\mathcal{D}^l$ (of weak augmentation) with the ground-truth and unlabeled data $\mathcal{D}^u$ (of strong augmentation) with the generated pseudo labels $\hat{\mathcal{Y}}$. We use standard random resize and random horizontal flip as the weak augmentation. Strong augmentation includes CutMix [27] and all data augmentation strategies used in the weak augmentation.

The overall loss consists of the supervised loss $\mathcal{L}_s$ and the unsupervised loss $\mathcal{L}_u$:

$$\mathcal{L}_s = \frac{1}{N_l} \sum_{i=1}^{N_l} \frac{1}{WH} \sum_{j=1}^{WH} \ell_{ce}(\boldsymbol{y}_{ij}, \boldsymbol{p}_{ij}), \tag{1}$$

$$\mathcal{L}_u = \frac{1}{N_u} \sum_{i=1}^{N_u} \frac{1}{WH} \sum_{j=1}^{WH} \ell_{ce}(\hat{\boldsymbol{y}}_{ij}, \boldsymbol{p}_{ij}), \tag{2}$$

where $\boldsymbol{p}_{ij}$ is the prediction of the $j$-th pixel in the $i$-th labeled (or unlabeled) image, $N_l$ and $N_u$ denote the number of labeled images and unlabeled images in a training batch, $W$ and $H$ represent the width and height of the input image, and $\ell_{ce}$ denotes the standard pixel-wise cross-entropy loss. We define the overall loss function as:

$$\mathcal{L} = \mathcal{L}_s + \alpha \mathcal{L}_u, \tag{3}$$

where $\alpha$ controls the contribution of the unsupervised loss.

## 3.3 Adaptive Equalization Learning

The baseline framework, though achieving competitive results compared with previous related works, neglects the key issues in semi-supervised semantic segmentation. Due to the limited labeled data, semi-supervised learning tends to have poor performance on some certain categories, e.g., tailed categories in Cityscapes dataset which exhibits a long-tailed label distribution. Insufficient training on these categories introduces more noise of pseudo labels which can disrupt the learning process. The proposed AEL framework aims to alleviate the degradation of under-performing categories during the semi-supervised training. Concretely, we maintain a confidence bank to record the performance of each category during training. The confidence bank enables us to identify the under-performing categories. To improve the performance of these categories and further make the training unbiased, we propose a series of technologies to efficiently leverage both labeled and unlabeled data, namely adaptive CutMix, adaptive Copy-Paste, adaptive equalization sampling and dynamic re-weighting.

**Confidence Bank.** To tackle the biased training, previous methods [25, 26, 22, 28] always rely on the prior knowledge such as the number of training samples of each category to design the ad hoc sampling and weighting strategies. However, the performance of each category is not always strictly proportional to the number of training samples, because some categories tend to have discriminative features and thus fewer samples are required for training. Inspired by the recent progress [29] which applies active learning on semantic segmentation, we propose to maintain a confidence bank to record the category-wise performance during training. An indicator is needed to assess the performance of each category.

We consider several indicators, namely *Confidence*, *Margin* and *Entropy*. Formally, we define *Confidence* indicator as:

$$\text{Conf}^c = \frac{1}{N_l} \sum_{i=1}^{N_l} \frac{1}{N_i^c} \sum_{j=1}^{N_i^c} p_{ij}^c, \ \ c \in \{1, \dots, C\} \tag{4}$$

where $C$ is the category number, $N_i^c$ denotes the number of pixels belonging to category $c$ according to its ground-truth $\boldsymbol{y}_i$, $p_{ij}^c$ denotes the $c$-th channel prediction of the $j$-th pixel in the $i$-th image.

Define *Margin* indicator as:

$$\text{Margin}^c = \frac{1}{N_l} \sum_{i=1}^{N_l} \frac{1}{N_i^c} \sum_{j=1}^{N_i^c} (p_{ij}^c - \max2_{c' \in \{1,\ldots,C\}} p_{ij}^{c'}), \ \ c \in \{1,\ldots,C\} \tag{5}$$

where $\max2(\cdot)$ denotes the second largest value operator. At last, we define *Entropy* indicator as:

$$\text{Ent}^c = -\frac{1}{N_l} \sum_{i=1}^{N_l} \frac{1}{N_i^c} \sum_{j=1}^{N_i^c} \sum_{c'=1}^{C} p_{ij}^{c'} \log p_{ij}^{c'}, \ \ c \in \{1,\ldots,C\}. \tag{6}$$

For all of the indicators, we only take into account predictions from labeled data. Experimentally, the confidence indicator serves best in our AEL and thus we adopt it by default (see Section 4.3 for the comparison). We use EMA to update the category-wise confidence at each training step:

$$\text{Conf}_k^c \leftarrow \tau \text{Conf}_{k-1}^c + (1-\tau)\text{Conf}_k^c, \ c \in \{1,\ldots,C\}, \tag{7}$$

where $k$ denotes the $k$-th iteration, $\tau \in [0,1)$ is the momentum coefficient which is set to $0.999$ experimentally. Through the confidence bank, we can easily identify the under-performing categories for the current model.

**Adaptive CutMix.** Here we introduce the proposed adaptive CutMix (see Figure 2 for illustration) which is applied on the unlabeled data. It aims to increase the frequency of occurrence of the under-performing samples from the unlabeled data. We first formulate the original CutMix [27] as:

$$\hat{I} = \text{CutMix}(\text{Crop}(I_1), I_2), \tag{8}$$

where $I_1$ and $I_2$ denote randomly selected unlabeled images, $\hat{I}$ is the augmented image, and $\text{Crop}(\cdot)$ represents the random crop operation.

Different from the original CutMix where unlabeled images are randmoly selected, the proposed adaptive CutMix gives under-performing categories a higher sampling probability. Specifically, we first convert the category-wise confidence stored in the confidence bank to the normalized sampling probability $r \in \mathbb{R}^C$, which can be formulated as:

$$r = \text{Softmax}(1 - \text{Conf}). \tag{9}$$

According to the sampling probability, we randomly select an unlabeled image containing the sampled category as $I_1$, and another unlabeled image from the training batch is randomly selected as $I_2$. The $\text{Crop}(\cdot)$ operation is performed on the region containing the chosen category. After that, we can generate the augmented image by Eq 8. Since the adaptive CutMix is performed on the unlabeled data without any annotations, we use predictions as approximate ground-truth, which works well in practice.

**Adaptive Copy-Paste.** Copy-Paste [30] is an effective data augmentation strategy for instance segmentation. It yields significant gains on the challenging LVIS benchmark [31], especially for rare object categories. The key idea behind the Copy-Paste augmentation is to paste objects from the source image to the target image. Inspired by this, we further propose the adaptive Copy-Paste (see Figure 2 for illustration) for semi-supervised semantic segmentation. Different from adaptive CutMix, adaptive Copy-Paste augmentation strives for efficiently leveraging the labeled data. Similarly, we involve confidence bank to assess category-wise performance and use Eq 9 to compute sampling probability. The under-performing categories have higher probability to be selected for Copy-Paste. Experimentally, the proposed adaptive Copy-Paste augmentation yields slightly better performance in the category level than the instance level. Thus we copy all pixels belonging to the sampled category in the source image and paste them on the target image. Following [30], the augmented image is composed of two randomly selected images from the labeled data and a large scale jittering is applied.

**Adaptive Equalization Sampling.** As described in Section 1, due to the limited and unbalanced labeled data, the training tends to be biased. To alleviate the training bias, we propose a novel adaptive equalization sampling strategy which focuses training on a sparse set of under-performing samples and prevents the vast number of well-trained samples from overwhelming the model during training. Concretely, we define the sampling rate $s^c$ for category $c$ as:

$$s^c = \left[ \frac{1 - \text{Conf}^c}{\max_{c \in \{1,\ldots,C\}}(1 - \text{Conf}^c)} \right]^\beta, \ c \in \{1,\ldots,C\}, \tag{10}$$

where $\beta$ denotes a tunable parameter. Instead of using all pixels to compute the unsupervised loss, for category $c$ with the sampling rate $s^c$, we randomly sample a subset of pixels according to their predictions. Then the unsupervised loss in Eq 2 can be reformulated as:

$$\mathcal{L}_u = \frac{1}{N_u} \sum_{i=1}^{N_u} \frac{1}{\sum_{j=1}^{WH} \mathbb{1}_{ij}} \sum_{j=1}^{WH} \ell_{ce}(\hat{\boldsymbol{y}}_{ij}, \boldsymbol{p}_{ij}) \mathbb{1}_{ij}, \tag{11}$$

where $\mathbb{1}_{ij} = 1$ indicates that the $j$-th pixel in the $i$-th image is sampled according to the sampling rate, otherwise $\mathbb{1}_{ij}$ is set to 0, the other terms are the same as in Eq 2.

**Dynamic Re-Weighting.** The performance of the model depends on the quality of pseudo labels. Existing methods [2, 4, 19] usually adopt a higher threshold on classification score to filter out most of the pixels with low-confidence. Though this strategy could alleviate the noise raised by pseudo-labeling, the strict criteria leads to lower recall for the pixels from under-performing categories, which hinders the training. Another option is to discard the threshold and involve all pixels into the training. However, much more noise is introduced simultaneously. To alleviate this issue, we propose a dynamic re-weighting strategy which adds a modulating factor to the unsupervised loss in the way of semi-supervised learning. On the basis of Eq 11, we formulate our final unsupervised loss as:

$$\mathcal{L}_u = \frac{1}{N_u} \sum_{i=1}^{N_u} \frac{1}{\sum_{j=1}^{WH} w_{ij}} \sum_{j=1}^{WH} w_{ij} \ell_{ce}(\hat{\boldsymbol{y}}_{ij}, \boldsymbol{p}_{ij}), \tag{12}$$

$$w_{ij} = \max_{c \in \{1,\ldots,C\}} (p_{ij}^c)^\gamma \mathbb{1}_{ij}, \tag{13}$$

where $\gamma$ is the tunable parameter. Different from the Focal Loss [32] where the modulating factor is used for reducing the loss contribution from easy samples, our formulation aims to allocate more contributions for the convincing samples. The combination of adaptive equalization sampling and dynamic re-weighting not only involves more samples from the under-performing categories into the training, but also alleviate the noise raised by pseudo-labeling.

## 4 Experiments

### 4.1 Setup

**Datasets.** Cityscapes [1] dataset is designed for urban scene understanding. It contains 30 classes and only 19 classes of them are used for scene parsing evaluation. The dataset contains $5,000$ finely annotated images and $20,000$ coarsely annotated images. The finely annotated $5,000$ images are split into $2,975$, $500$ and $1,525$ images for training, validation and testing respectively.

PASCAL VOC 2012 [33] dataset is a standard object-centric semantic segmentation dataset. It contains 20 foreground object classes and a background class. The strand training, validation and testing sets consist of $1,464$, $1,449$ and $1,556$ images, respectively. Following common practice, we use the augmented set [34] which contains $10,582$ images as the training set.

ADE20K dataset [35] is a large scale scene parsing benchmark which contains dense labels of 150 stuff/object categories. The dataset includes 20K/2K/3K images for training, validation and testing.

For both Cityscapes and PASCAL VOC 2012 datasets, 1/2, 1/4, 1/8, 1/16 and 1/32 training images are randomly sampled as the labeled training data, and the remaining images are used as the unlabeled data. For each protocol, AEL provides 5 different data folds and the final performance is the average of 5 folds. In addition, we also evaluate our method on the setting where the full Cityscapes `train` set is used as the labeled data and $1,000$, $3,000$ and $5,000$ images and randomly selected from the Cityscapes `coarse` set as the unlabeled data.

**Evaluation.** We use single scale testing and adopt mean of Intersection over Union (mIoU) as the metric to evaluate the performance. We report the results on the Cityscapes `val` set and PASCAL VOC 2012 `val` set in comparisons with state-of-the-art methods. All ablation studies are conducted on the Cityscapes `val` set under 1/16 and 1/32 partition protocols.

**Implementation Details.** We use ResNet-101 pretrained on ImageNet [36] as our backbone, remove the last two down-sampling operations and employ dilated convolutions in the subsequent convolution layers, making the output stride equal to 8. We use DeepLabv3+ [37] as the segmentation head. For

Table 1: Comparison with state-of-the-art methods on the **Cityscapes** `val` set under different partition protocols. All the methods are based on DeepLabv3+ with ResNet-101 backbone.

| Method | 1/32 (93) | 1/16 (186) | 1/8 (372) | 1/4 (744) | 1/2 (1488) |
|---|---|---|---|---|---|
| Supervised | 57.89 | 62.96 | 69.81 | 74.23 | 77.46 |
| MT [11] | 64.07 | 68.05 | 73.56 | 76.66 | 78.39 |
| CCT [4] | 66.35 | 69.32 | 74.12 | 75.99 | 78.10 |
| Cutmix-Seg [2] | 69.11 | 72.13 | 75.83 | 77.24 | 78.95 |
| GCT [19] | 63.21 | 66.75 | 72.66 | 76.11 | 78.34 |
| AEL (Ours) | **74.28** | **75.83** | **77.90** | **79.01** | **80.28** |

Table 2: Comparison with state-of-the-art methods on the **PASCAL VOC 2012** `val` set under different partition protocols. All the methods are based on DeepLabv3+ with ResNet-101 backbone.

| Method | 1/32 (331) | 1/16 (662) | 1/8 (1323) | 1/4 (2646) | 1/2 (5291) |
|---|---|---|---|---|---|
| Supervised | 70.14 | 70.60 | 73.12 | 76.35 | 77.21 |
| MT [11] | 70.56 | 71.29 | 73.33 | 76.61 | 78.08 |
| CCT [4] | 71.22 | 71.86 | 73.68 | 76.51 | 77.40 |
| Cutmix-Seg [2] | 73.39 | 73.56 | 73.96 | 77.58 | 78.12 |
| GCT [19] | 70.32 | 70.90 | 73.29 | 76.66 | 77.98 |
| AEL (Ours) | **76.97** | **77.20** | **77.57** | **78.06** | **80.29** |

Cityscapes dataset, we use stochastic gradient descent (SGD) optimizer with initial learning rate 0.01, weight decay 0.0005 and momentum 0.9. Moreover, we adopt the 'poly' learning rate policy, where the initial learning rate is multiplied by $(1 - \frac{iter}{max\_iter})^{0.9}$. We adopt the crop size as $769 \times 769$, batch size as 16 and training iterations as 18k. For PASCAL VOC 2012 dataset, we set the initial learning rate as 0.001, weight decay as 0.0001, crop size as $513 \times 513$, batch size as 16 and training iterations as 30k. We use random horizontal flip and random resize as the default data augmentation if not specified. All the supervised baselines are trained on the labeled data.

### 4.2 Comparison with State-of-the-Art Methods

We compare our method with recent semi-supervised semantic segmentation methods, including Mean Teacher (MT) [11], Cross-Consistency Training (CCT) [4], Guided Collaborative Training (GCT) [19] and Cutmix-Seg [2]. For a fair comparison, we re-implement all above methods and adopt the same network architecture (DeepLabv3+ with ResNet-101 backbone).

**Results on Cityscapes Dataset.** Table 1 compares AEL with state-of-the-art methods on the Cityscapes `val` set. Without leveraging any unlabeled data, the performance of the supervised baseline is unsatisfactory under various data partition protocols, especially for the fewer data settings, e.g., 1/32 and 1/16 protocols. Our method consistently promotes the baseline, achieving the improvements of +16.4%, +12.9%, +8.1%, +4.8% and +2.8% under 1/32, 1/16, 1/8, 1/4 and 1/2 partition protocols respectively. Our method also significantly outperforms the existing state-of-the-art methods by a large margin under all data partition protocols. In particular, AEL outperforms the existing best method Cutmix-Seg by +5.2% under extremely few data setting (1/32 protocol), and surpasses Cutmix-Seg by +1.3% under the 1/2 protocol.

**Results on PASCAL VOC 2012 Dataset.** Table 2 shows comparison with state-of-the-art methods on the PASCAL VOC 2012 `val` dataset. AEL achieves consistent performance gains over the supervised baseline, obtaining an improvements of +6.8%, +7.0%, +4.1%, +1.7% and +3.1% under 1/32, 1/16, 1/8, 1/4 and 1/2 partition protocols respectively. We can see that over all protocols, AEL outperforms the state-of-the-art methods. For example, our method outperforms the previous best method by +3.6% and +2.2% under the 1/32 and 1/2 partition protocols.

Table 3: Ablation study on the effectiveness of different components: Dynamic Re-weighting (DR), Adaptive Equalization Sampling(AES), Adaptive CutMix (ACM), Adaptive Copy-Paste (ACP).

| DR | AES | ACM | ACP | 1/32 (93) | 1/16 (186) |
|---|---|---|---|---|---|
| | | | | 69.11 | 72.13 |
| ✓ | | | | 70.27 | 73.85 |
| | ✓ | | | 71.65 | 74.12 |
| | | ✓ | | 70.49 | 73.89 |
| | | | ✓ | 69.69 | 72.64 |
| ✓ | ✓ | | | 72.51 | 74.39 |
| ✓ | ✓ | ✓ | | 73.43 | 75.12 |
| ✓ | ✓ | ✓ | ✓ | **74.28** | **75.83** |

## 4.3 Ablation Study

To further understand the advantages of AEL, we conduct a series of ablation studies that examine the effectiveness of different components and different hyper-parameters. All experiments are conducted on the validation set of Cityscapes dataset.

**The Effectiveness of Different Components.** We ablate each component of AEL step by step. Table 3 reports the studies. We use the basic framework described in Section 3.2 as our baseline, which achieves 69.11% and 72.13% under 1/32 and 1/16 protocols respectively. We first evaluate the effectiveness of each single component. As shown in the table, Dynamic Re-weighting (DR) improves the baseline by +1.1% and +1.7% under 1/32 and 1/16 partition protocols. Adaptive Equalization Sampling (AES) alleviates the biased training issue, achieving the improvements of +2.5% and +2.0% over the baseline. Adaptive CutMix (ACM) and Adaptive Copy-Paste (ACP) data augmentation approaches give more chance for under-performing categories to be sampled, and bring the improvements of +1.3% / +1.7% and +0.5% / +0.5% respectively. Furthermore, we present the performance gains in a progressive manner. On top of the DR, by leveraging AES strategy on the unsupervised loss, our method obtains improvements of +2.3% and +0.5% under 1/32 and 1/16 protocols. The two proposed data augmentation approaches further boost the performance to 74.28% and 75.83%, demonstrating the effectiveness of our adaptive learning.

**Ablation Study on Hyper-Parameters.** Table 5 ablates the tunable parameter $\gamma$ in dynamic re-weighting (in Eq 13), where $\gamma = 2$ yields slightly better performance. Dynamic re-weighting is found to be insensitive to $\gamma$.

Table 6 ablates the influence of different indicators, including Confidence (in Eq 4), Margin (in Eq 5), and Entropy (in Eq 6). We use the Confidence as the default indicator to assess the category-wise performance during training due to its best performance.

Adaptive CutMix requires a criteria to identify whether an unlabeled image contains a certain class. We use the ratio between pseudo labels of a certain category and total pixels of the input image as the criteria. Table 7 ablates different ratios.

Table 8 studies the number of sampled categories $K$ in the Adaptive Copy-Paste. We find that $K = 3$ achieves the best performance. One potential reason is that a smaller $K$ provides less training samples from the under-performing categories while a larger $K$ may increase the difficulty for training.

Table 9 ablates the loss weight $\alpha$ which is used to balance the supervised loss and unsupervised loss as shown in Eq 3. As illustrated in the table, $\alpha = 1$ achieves the best performance. We use $\alpha = 1$ in our approach for all the experiments.

## 4.4 Per-class Results

Since the class imbalance problem is severe in the Cityscapes dataset, we provide per-class results under 1/32 data partition protocol in Table 4. We choose 9 classes with the least training samples in the Cityscapes dataset as tail classes, i.e. wall, traffic light, traffic sign, rider, truck, bus, train, motorcycle and bicycle. As shown in the table, our method not only achieves the best overall mIoU,

Table 4: Per-class results on **Cityscapes** `val` set under 1/32 data partition protocol. All the methods are based on DeepLabv3+ with ResNet-101 backbone.

| Methods | mIoU | mIoU_tail | road | sidewalk | building | fence | pole | vegetation | terrain | sky | person | car | wall | traffic light | traffic sign | rider | truck | bus | train | motorcycle | bicycle |
|---|---|---|---|---|---|---|---|---|---|---|---|---|---|---|---|---|---|---|---|---|---|
| | | | Head Classes | | | | | | | | | | Tail Classes | | | | | | | | |
| Supervised | 57.9 | 39.8 | 94.6 | 72.5 | 87.4 | 42.4 | 51.6 | 88.3 | 49.8 | 91.1 | 74.5 | 89.4 | 21.7 | 47.7 | 59.1 | 33.7 | 43.3 | 37.2 | 11.4 | 42.1 | 62.2 |
| GCT [19] | 63.2 | 48.1 | 96.9 | 75.8 | 89.8 | 40.3 | 57.5 | 91.1 | 53.5 | 93.1 | 78.1 | 91.6 | 23.6 | 58.9 | 70.1 | 43.4 | 25.8 | 45.7 | 49.2 | 45.0 | 71.4 |
| MT [11] | 64.1 | 50.4 | 96.7 | 75.6 | 89.5 | 40.0 | 57.3 | 91.0 | 53.2 | 92.80 | 77.9 | 91.3 | 26.2 | 61.1 | 72.3 | 45.8 | 28.0 | 48.1 | 51.6 | 47.1 | 73.8 |
| CCT [4] | 66.4 | 54.2 | 95.7 | 77.2 | 88.6 | 46.5 | 58.5 | 90.1 | 55.5 | 91.5 | 77.9 | 91.8 | 27.9 | 60.5 | 71.8 | 48.0 | 44.5 | 61.4 | 50.7 | 52.0 | 70.5 |
| Cutmix-Seg [2] | 69.1 | 58.7 | **97.2** | 78.6 | 90.1 | 48.1 | 60.1 | 91.5 | 57.2 | 93.0 | 79.6 | 93.3 | 32.4 | 64.8 | 76.5 | 52.3 | 49.4 | 66.0 | 54.8 | 56.7 | 75.1 |
| AEL (Ours) | **74.3** | **67.9** | 97.1 | 78.7 | 90.3 | 52.3 | 62.0 | 91.7 | 59.2 | 93.8 | 81.6 | 94.0 | 37.3 | 67.9 | 77.6 | 60.5 | 65.6 | 83.8 | 74.3 | 66.9 | 77.0 |

Table 5: Study on $\gamma$ of dynamic re-weighting.

| $\gamma$ | 1/32 | 1/16 |
|---|---|---|
| 0 | 69.11 | 72.13 |
| 0.5 | 69.74 | 73.28 |
| 1 | 69.35 | 73.67 |
| **2** | **70.27** | **73.85** |
| 3 | 70.26 | 73.40 |

Table 6: Study on different indicators for AES.

| Indicator | 1/32 | 1/16 |
|---|---|---|
| None | 70.27 | 73.85 |
| Ent | 71.38 | 73.21 |
| **Conf** | **72.51** | **74.39** |
| Margin | 70.86 | 73.05 |

Table 7: Study on different ratios in ACM.

| Ratio | 1/32 | 1/16 |
|---|---|---|
| 0.001 | 73.27 | 74.28 |
| 0.003 | 73.29 | 74.36 |
| **0.005** | **73.43** | **75.12** |
| 0.01 | 72.78 | 73.66 |

but also obtains significant improvements on tail classes. In particular, Our method outperforms the existing best method Cutmix-Seg by +5.2% in overall mIoU and +9.2% in mIoU for tail classes under 1/32 partition protocol.

### 4.5 Performance on the Full Labeled Set

We conduct experiments where the full Cityscapes `train` set is used as the labeled dataset and the Cityscapes `coarse` set is used as the unlabeled dataset. We do not leverage any annotations from the `coarse` set though it provides coarsely annotated ground-truth. We randomly sample 1,000, 3,000 and 5,000 images from the `coarse` set to verify the proposed method. As shown in Table 10, the proposed AEL can still improve the supervised baselines by leveraging the unlabeled data though a large amount of labeled data is provided.

### 4.6 Results on ADE20K Dataset

We further provide results on the ADE20K dataset [35]. Since no previous methods in semi-supervised segmentation conducted experiments on ADE20K dataset, we compare our method with supervised baseline and existing best method Cutmix-Seg on the dataset. As shown in Table 11, our method consistently promotes the supervised baseline by 6.36%, 5.70%, 5.67%, 3.18% and 1.45%, and outperforms the Cutmix-Seg by 2.25%, 3.38%, 2.48%, 1.31% and 1.26% under 1/32, 1/16, 1/8, 1/4 and 1/2 partition protocols, respectively.

### 4.7 Qualitative Results

Figure 3 shows the visualization results of different methods evaluated on the Cityscapes `val` set. We compare the proposed AEL with ground-truth, supervised baseline and our basic framework described in Section 3.2. Benefiting from a series of technologies designed for the balanced training, AEL achieves great performance on not only head categories (e.g. *Road*), but also tailed categories (e.g. *Rider* and *Bicycle*).

Table 8: Study on number of sampled categories $K$ in ACP.

| $K$ | 1/32 | 1/16 |
|---|---|---|
| 1 | 72.18 | 74.85 |
| 2 | 72.84 | 74.95 |
| **3** | **74.28** | **75.83** |
| 4 | 73.43 | 74.10 |

Table 9: Study on loss weight $\alpha$.

| $\alpha$ | 1/32 | 1/16 |
|---|---|---|
| 0.5 | 71.85 | 74.61 |
| **1.0** | **74.28** | **75.83** |
| 1.5 | 74.10 | 73.44 |
| 2.0 | 73.79 | 72.86 |

Table 10: Performance on the full Cityscapes `train` set.

| Number | Baseline | AEL |
|---|---|---|
| 0 | 80.16 | - |
| 1000 | 80.22 | **80.28** |
| 3000 | 80.55 | **81.36** |
| 5000 | 80.92 | **81.95** |

Table 11: Comparison with supervised baseline and Cutmix-Seg on the **ADE20K** `val` set under different partition protocols. All the methods are based on DeepLabv3+ with ResNet-101 backbone.

| Method | 1/32 (631) | 1/16 (1263) | 1/8 (2526) | 1/4 (5052) | 1/2 (10105) |
|---|---|---|---|---|---|
| Supervised | 22.04 | 27.52 | 32.36 | 36.39 | 41.97 |
| Cutmix-Seg [2] | 26.15 | 29.84 | 35.55 | 38.26 | 42.16 |
| AEL (Ours) | **28.40** | **33.22** | **38.03** | **39.57** | **43.42** |

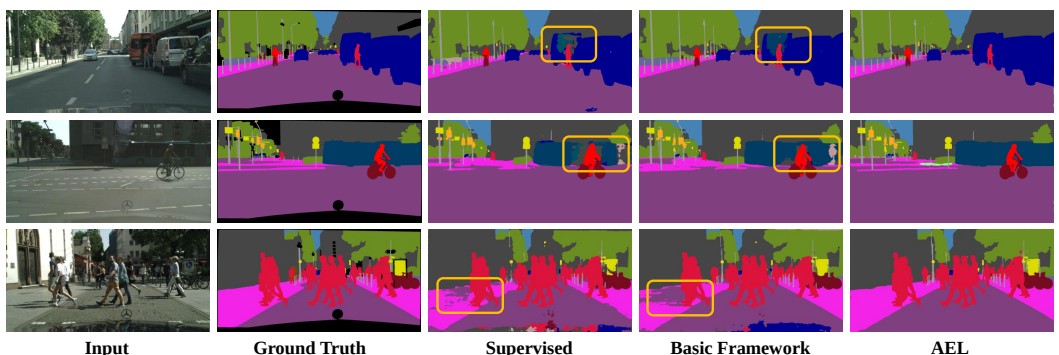

**Input**    **Ground Truth**    **Supervised**    **Basic Framework**    **AEL**

Figure 3: Qualitative results on the Cityscapes `val` set. From left to right: input image, ground-truth, predictions of the supervised baseline, predictions of our basic framework and predictions of the proposed AEL. Orange rectangles highlight the unsatisfactory segmentation results.

## 5   Conclusion

In this paper, we propose a novel Adaptive Equalization Learning (AEL) framework for semi-supervised semantic segmentation. Different from the existing methods dedicating to the design of consistency regularization or pseudo-labeling, AEL aims to adaptively balance the training based on the fact that pixel categories in common semantic segmentation datasets tend to be imbalanced. We introduce a confidence bank to dynamically record the category-wise performance at each training step, which enables us to identify the under-performing categories and adaptively tilt training towards these categories. Several technologies are proposed to make the training unbiased, namely adaptive Copy-Paste and CutMix, adaptive equalization sampling and dynamic re-weighting. Through the adaptive design, AEL outperforms the state-of-the-art methods by a large margin on the Cityscapes and Pascal VOC benchmarks under various data partition protocols.

## Acknowledgment

This work was supported by the National Key R&D Program of China under grant 2017YFB1002804 and National Natural Science Foundation of China (No. 31771230).

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
