## A   Implementation Details of Adaptive CutMix

Adaptive CutMix requires a randomly selected unlabeled image that contains the sampled category, thus we need a criteria to determine whether an unlabeled image $I$ contains the category $c$ ($c \in \{1, \ldots, C\}$). Concretely, in our implementation, we maintain a dictionary which is dynamically updated during training to record the category-wise information for each unlabeled image. For each unlabeled image from a training batch, we use the teacher model to generate pseudo labels as the approximate ground-truth. Then we compute the ratio $r^c$ between pseudo labels of the category $c$ and the total pixels of the image $I$. With a predefined threshold $r^*$, we can easily identify the category $c$ is existing in the image $I$ if $r^c > r^*$. Notice an unlabeled image may be identified as containing multi categories. Table 6 in Section 4.3 ablates the threshold $r^*$.

## B   Hyper-parameters for PASCAL VOC 2012 Dataset

In this section, we present hyper-parameter settings of our proposed AEL on PASCAL VOC 2012 dataset [33]. All hyper-parameters are chosen carefully with extensive experiments. Concretely, tunable parameter $\gamma$ is set as 2 and confidence is used as the indicator to assess the category-wise performance during training. The ratio $r^*$ in adaptive CutMix is set as 0.03 and the number of sampled categories $K$ is set as 1 since the image in PASCAL VOC 2012 dataset contains fewer categories and larger instances than Cityscapes dataset [1]. The loss weight $\alpha$ is also set as 1 as in Cityscapes dataset.

## C   Qualitative Results on PASCAL VOC 2012 Dataset

Figure 4 shows the visualization results on the PASCAL VOC 2012 [33] val set. We compare the proposed AEL with ground-truth, supervised baseline and our basic framework described in Section 3.2. AEL achieves promising visual quality and further improves fine details.

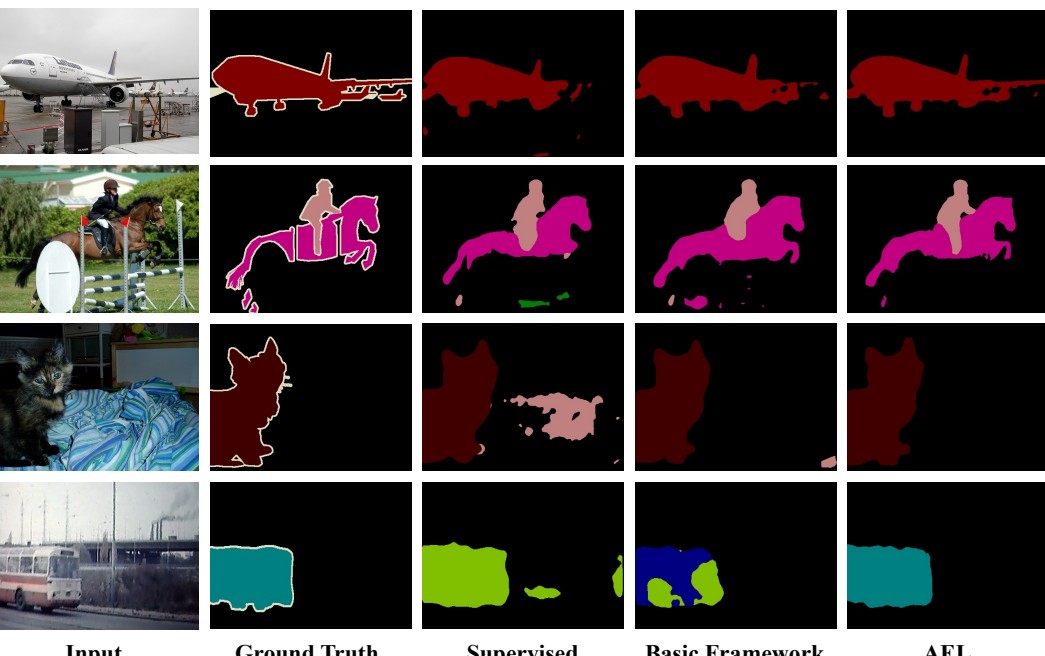

|          Input          Ground Truth          Supervised          Basic Framework          AEL |

Figure 4: Visualization results on the PASCAL VOC 2012 val set. From left to right: input image, ground-truth, predictions of the supervised baseline, predictions of our basic framework and predictions of the proposed AEL.