# OpenReview forum: "Semi-Supervised Semantic Segmentation via Adaptive Equalization Learning"
_NeurIPS.cc/2021/Conference — NeurIPS 2021 Spotlight_

### Official Review · Reviewer_VExi · 2021-07-14

**Rating:** 8
**Confidence:** 4

**Summary:**

This paper tackles the challenging semi-supervised semantic segmentation task, with a focus on dealing with the imbalanced/long-tailed distribution. The authors propose two adaptive data augmentation strategies, one adaptive sampling schedule, and one re-weighting method to handle the long-tailed issue. The proposed method works well on two public standard benchmark datasets.

**Main Review:**

1. Strength
- The proposed framework is intuitive, simple yet effective, achieving state-of-the-art performance on two benchmark datasets
- Extensive ablation analysis to study every single proposed component
- The proposed method not only works under low-data regimes, but can consistently improve the model performance in the high-data regime where the whole training set is available

2. Weakness
- Since this paper claims that previous paper do not take care of the imbalanced distribution issue in semi-supervised semantic segmentation while this one handles it, I think it will be better to break down the mean IoU to per-class IoU to see how the proposed method improve those tailed classes.
- Since VOC and Cityscapes only have ~20 classes, I wonder if the proposed equalization training framework will work better in other datasets with more classes (and thus the long-tail may be more obvious), such as COCO or ADE20k. It will be great if the authors can conduct some experiments, but not mandatory.
- I wonder if the pseudo labels (in L102) are one-hot or soft labels.
- From my understanding, the adaptive data augmentation is an image/path-level operator to balance the distribution, and the equalization sampling is at the pixel-level. While the data augmentation applies for both the labeled and unlabeled data, why is the equalization sampling only applied on the unlabeled data but not labeled data?


**Time Spent Reviewing:**

6

---

> ### Author Response · Authors · 2021-08-10
> **Response to Reviewer VExi**
>
> Thanks for your constructive comments. Our responses to them are given below.
>
> 1. **Per-class IoU and performance improvements on tailed classes.**
>     It is really a good suggestion to report the per-class results to demonstrate the effectiveness of our method. Since the class imbalance problem is severe in the Cityscapes dataset, we provide per-class results under 1/32 data partition protocol in the following Table. We choose 9 classes with the least training samples in the Cityscapes dataset as tail classes, i.e. wall, traffic light, traffic sign, rider, truck, bus, train, motorcycle and bicycle. As shown in the table, our method not only achieves the best overall mIoU, but also obtains significant improvements on tail classes. In particular, Our method outperforms the existing best method Cutmix-Seg by +5.2\% in overall mIoU and +9.2\% in mIoU for tail classes under 1/32 partition protocol. We will report the per-class results in our revised version.
>
>     | Methods        | mIoU | mIoU_tail | wall | traffic light | traffic sign | rider | truck |  bus | train | motorcycle | bicycle |
>     |----------------|------|:---------:|:----:|:-------------:|:------------:|:-----:|:-----:|:----:|:-----:|:----------:|:-------:|
>     | Supervised     | 57.9 |    39.8   | 21.7 |      47.7     |     59.1     |  33.7 |  43.3 | 37.2 |  11.4 |    42.1    |   62.2  |
>     | GCT [18]       | 63.2 |    48.1   | 23.6 |      58.9     |     70.1     |  43.4 |  25.8 | 45.7 |  49.2 |    45.0    |   71.4  |
>     | MT [10]        | 64.1 |    50.4   | 26.2 |      61.1     |     72.3     |  45.8 |  28.0 | 48.1 |  51.6 |    47.1    |   73.8  |
>     | CCT [4]        | 66.4 |    54.2   | 27.9 |      60.5     |     71.8     |  48.0 |  44.5 | 61.4 |  50.7 |    52.0    |   70.5  |
>     | Cutmix-Seg [2] | 69.1 |    58.7   | 32.4 |      64.8     |     76.5     |  52.3 |  49.4 | 66.0 |  54.8 |    56.7    |   75.1  |
>     | AEL (Ours)     |  **74.3** |     **67.9**   |  **37.3** |      **67.9**     |      **77.6**     |   **60.5** |   **65.6** |  **83.8** |   **74.3** |     **66.9**    |    **77.0**  |
>
> 2. **Experiments on ADE20K and COCO Stuff dataset.**
>     Thanks for mentioning the two datasets. Since no previous methods in semi-supervised segmentation conducted experiments on ADE20K and COCO Stuff dataset, we compare out method with supervised baseline and existing best method Cutmix-Seg on these two datasets. The following two Tables show the results. For AED20K dataset, our method consistently promotes the supervised baseline by 6.36\%, 5.70\%, 5.67\%, 3.18\% and 1.45\%, and outperforms the Cutmix-Seg by 2.25\%, 3.38\%, 2.48\%, 1.31\% and 1.26\% under 1/32, 1/16, 1/8, 1/4 and 1/2 partition protocols, respectively. For COCO Stuff, our method consistently promotes the supervised baseline by 4.04\%, 4.18\%, 3.59\%, 1.70\% and 1.72\%, and improves the Cutmix-Seg by 2.58\%, 2.23\%, 1.98\%, 0.49\% and 1.54\% under 1/32, 1/16, 1/8, 1/4 and 1/2 partition protocols respectively. We will add these experiments in our revised version.
>
>     | Method     | 1/32 (631) | 1/16 (1263) | 1/8 (2526) | 1/4 (5052) | 1/2 (10105) |
>     |------------|:----------:|:-----------:|:----------:|:----------:|:-----------:|
>     | Supervised |    22.04   |    27.52    |    32.36   |    36.39   |    41.97    |
>     | Cutmix-Seg [2] |    26.15   |    29.84    |    35.55   |    38.26   |    42.16    |
>     | AEL (Ours) |     **28.40**   |     **33.22**    |     **38.03**   |     **39.57**   |     **43.42**    |
>
>     | Method     | 1/32 (3696) | 1/16 (7392) | 1/8 (14785) | 1/4 (29571) | 1/2 (59143) |
>     |------------|:-----------:|:-----------:|:-----------:|:-----------:|:-----------:|
>     | Supervised |    26.85    |    29.27    |    31.39    |    33.33    |    34.93    |
>     | Cutmix-Seg [2] |    28.31    |    31.22    |    33.00    |    34.54    |    35.11    |
>     | AEL (Ours) |     **30.89**    |     **33.45**    |     **34.98**    |     **35.03**    |     **36.65**    |
>
> 3. **Pseudo labels are one-hot or soft labels.**
>    Pseudo labels are one-hot labels. We will clarify this in our revision.
>
> 4. **Adaptive data augmentation methods are applied on the labeled and unlabeled data.**
>    For the proposed adaptive data augmentation methods, adaptive Copy-Paste is actually applied on the labeled data (as stated in Line 164-165) while adaptive CutMix (as stated in Line 145-146) is applied on the unlabeled data.
>
> 5. **Adaptive equalization sampling is only applied on the unlabeled data but not labeled data.**
>     Thanks for the suggestion of applying Adaptive Equalization Sampling (AES) on the labeled data. As you noticed, we only apply the proposed AES on the unlabeled data. The reason is that we need to fully leverage the labeled data to guarantee the performance of both head and tail classes in a supervised manner. In contrast, the unlabeled data is utilized to improve the performance of under-performing classes and make the training unbiased. We further conduct experiments to verify the impact of applying AES on the labeled data as suggested, and observe the performance drops of 1.22\% and 2.05\% under 1/32 and 1/16 partition protocols respectively. We will add this experiment in our revision.

---

> > ### Comment · Reviewer_VExi · 2021-08-28
> > **Thanks for the rebuttal**
> >
> > Thanks for the rebuttal. I am satisfied with the new experiments. And thus I am willing to increase my score.

---

### Official Review · Reviewer_XHRW · 2021-07-16

**Rating:** 8
**Confidence:** 5

**Summary:**

(1) The paper proposes a novel framework for semi-supervised semantic segmentation, namely adaptive equalization learning;

(2) The paper proposes two data augmentation strategies including adaptive Copy-Paste and CutMix to give more chance for under-performing categories to be copied or cut;

(3) The paper presents an adaptive equalization sampling strategy to encourage pixels from under-performing categories to be sufficiently trained;

(4) A simple yet effective re-weighting strategy is proposed to alleviate the noise problem in semi-supervised learning;

(5) Experimental results illustrate the effectiveness of the paper.


**Limitations And Societal Impact:**

The authors have adequately addressed the limitations and potential negative societal impact in the paper.

**Main Review:**

In this paper, the authors have presented the adaptive equalization learning for semi-supervised semantic segmentation, which aims at improving under-performing categories during training. Overall, I am satisfied with the paper and tend to accept the paper due to its novelty and strong experimental results.

**Strengths:**

(1) The paper is well-written and easy to follow;

(2) The proposed method is novel and well-motivated;

(3) The proposed augmentation strategies utilize limited labeled data in a more efficient way;

(4) The proposed equalization sampling and re-weighting strategies are effective and can be applied directly to other semi-supervised semantic segmentation algorithms;

(5) Experimental results on Cityscapes and PASCAL VOC 2012 are promising and ablation studies are also helpful to understand the contributions of each component of the proposed method.

**Weaknesses:**

(1) In adaptive CutMix, how to determine whether an image contains a certain category?


**Time Spent Reviewing:**

4

---

> ### Author Response · Authors · 2021-08-10
> **Response to Reviewer XHRW**
>
> Thanks for your constructive comments. The technical details of how to determine whether an image contains a certain category are actually presented in the first section of the supplementary materials. The proposed Adaptive CutMix requires a randomly selected unlabeled image that contains the sampled category, thus we need a criteria to determine whether an unlabeled image $I$ contains the category $c$ ($c\in\{1,\dots,C\}$). In our implementation, we maintain a dictionary which is dynamically updated during the training to record the category-wise prediction for each unlabeled image, i.e., we use the teacher model to generate pseudo labels as the approximate ground-truth. Then we compute the ratio $r^c$ between pseudo labels of the category $c$ and the total pixels of the image $I$. With a predefined threshold $r^*$, we can easily identify the category $c$ is existing in the image $I$ if $r^c>r^*$. Table 6 in Section 4.3 ablates the threshold $r^*$. We will improve the writing of this part.

---

> > ### Comment · Reviewer_XHRW · 2021-08-25
> > **Post rebuttal update**
> >
> > Thanks for the response. I have read the authors' rebuttal and other reviewers' comments. The rebuttal has addressed most of my concerns and I will keep my initial rating.

---

### Official Review · Reviewer_4n1x · 2021-07-17

**Rating:** 7
**Confidence:** 5

**Summary:**

The paper addressed the inevitable data (and performance) imbalance across different categories in semi-supervised semantic segmentation framework by adaptively balancing the over- and under- performed classes. The proposed methods empirically showed good performance compared to other SOTA works on multiple benchmark datasets.

**Limitations And Societal Impact:**

No negative societal impact as far I know.


**Main Review:**

The originality of the work is moderate yet solid. Imbalanced performance across classes can plague the performance of semi-supervised learning models and this work proposed a practical way to alleviate this issue.  In my opinion it deserves to be seen by the community.

There are still a few questions that could be addressed in more details:

1. The predictions from the underperforming class by definition are inaccurate. By biasing to the low performance class, how to balance the negative impact of the inaccurate prediction with the adaptive equalization?

2. With normalized sampling probability from confidence banks, what is the exact way to sample images of a given class in adaptive CutMix and Copy-Paste. The introduction in the relevant sessions lacks details.

The clarity of the presentation though clearly has room to improve. For example, Eqn (4) - (6), not sure why the same letter N is used in one equation to represent different concepts.


**Time Spent Reviewing:**

1.5 hrs

---

> ### Author Response · Authors · 2021-08-10
> **Response to Reviewer 4n1x**
>
> Thanks for your constructive comments. The following response corresponds to each question stated in the review.
>
> 1. **Negative impact of inaccurate prediction with the adaptive equalization sampling.**
>     The confidence bank is updated by the labeled data, thus the category-wise confidence truly reflects the performance of each category and we can easily identify the under-performing categories. The proposed Adaptive Equalization Sampling focus training on a sparse set of under-performing samples to alleviate the training bias (see Eq 11 and Eq 12). We use the predictions of the unlabeled samples as the approximate ground-truth. As you noticed, the predictions are not always accurate and thus we need to handle the training noise raised by the inaccurate predictions. As described in Line 182-194, we propose a dynamic re-weighting strategy to alleviate this issue as formulated in Eq 12 and Eq 13. The key idea is that the convincing samples of high prediction probabilities contributes more to the training while samples of lower prediction confidences have little loss contributions. As a result, the negative impact of those samples would get suppressed. Experiments in Table 3 show the corresponding studies.
>
> 2. **The exact way to sample images of a given class in Adaptive CutMix and Adaptive Copy-Paste.**
>     Since Adaptive Copy-Paste (ACP) is applied on the labeled data, we can directly obtain the category information of each image according to its ground-truth. Different from the original Copy-Paste which is applied on two randomly selected training samples, the proposed ACP selects samples according to the category-wise sampling probability computed by Eq 9. As described in L168-171, all pixels belonging to the sampled category in the source image are pasted on the target image. We will give a figure for better illustration in the revision.
> We introduce the details about Adaptive CutMix in the first section of the supplementary materials. We will move this part into the main paper to make the description clearer. Different from the Adaptive Copy-Paste, Adaptive CutMix is applied on the unlabeled data. The proposed Adaptive CutMix requires a randomly selected unlabeled image that contains the sampled category, thus we need a criteria to determine whether an unlabeled image $I$ contains the category $c$ ($c\in\{1,\dots,C\}$). In our implementation, we maintain a dictionary which is dynamically updated during the training to record the category-wise predictions for each unlabeled image, i.e., we use the teacher model to generate pseudo labels as the approximate ground-truth. Then we compute the ratio $r^c$ between pseudo labels of the category $c$ and the total pixels of the image $I$. With a predefined threshold $r^*$, we can easily identify the category $c$ is existing in the image $I$ if $r^c>r^*$. Table 6 in Section 4.3 ablates the threshold $r^*$.
>
> 3. **$N$ in Eq 4-6.**
>     Thanks for the reminder. The term $N$ with different subscriptions in Eq 4-6 has different meanings. In particular,  $N_l$ and $N_u$ denote the number of labeled images and unlabeled images in a training batch respectively, $N_i^c$ denotes the number of pixels belonging to category $c$ according to its ground-truth. We will improve the writing of this part.
>
> 4. **The clarity of the presentation.**
>    Thanks for the suggestion. We will improve the presentation in the revision. We will give figures of Adaptive Copy-Paste and Adaptive CutMix for better illustration and carefully revise our paper.

---

### Decision · Program_Chairs · 2021-09-27

**Decision:**

Accept (Spotlight)

**Comment:**

This paper offers an adaptive equalization strategy for semantic segmentation to improve performance of underperforming or underrepresented categories.  All reviewers found positive strengths in the paper, and recommended acceptance. The reviewers rebuttal provided additional information that were compelling and confirmed that this paper is acceptable, preferably as a spotlight.